# Intervention Strategies to Overcome HPV Vaccine Hesitancy Among Hispanic Immigrants in the USA: A Video-Based Approach

**DOI:** 10.3390/vaccines13060574

**Published:** 2025-05-28

**Authors:** Isaiah Aduse-Poku, Diego A. Ardon, Alexis B. Call, Spencer C. Davis, Preston Evans, Spencer Johanson, Ruth J. Larson, James Rencher, Isaac A. Woolley, Brian D. Poole, Jamie L. Jensen

**Affiliations:** 1Department of Biology, Brigham Young University, Provo, UT 84602, USA; aduse@student.byu.edu (I.A.-P.); dardon@student.byu.edu (D.A.A.); acall10@student.byu.edu (A.B.C.); pe180797@student.byu.edu (P.E.); spenc100@student.byu.edu (S.J.); iwoolley@student.byu.edu (I.A.W.); 2Department of Microbiology and Molecular Biology, Brigham Young University, Provo, UT 84602, USA; sdavis45429@gmail.com (S.C.D.); brian_poole@byu.edu (B.D.P.); 3Department of Public Health, Brigham Young University, Provo, UT 84602, USA; ruthjamisonlarson@gmail.com; 4Department of Mathematics, Brigham Young University, Provo, UT 84602, USA; jamesgr1@student.byu.edu

**Keywords:** HPV vaccine, vaccine hesitancy, Hispanic immigrants

## Abstract

Background/Objectives: Hispanic immigrants (HIs) in the U.S.A. are disproportionately affected by cervical cancer compared to other groups, at least partly due to low HPV vaccination rates. The aim of this study was to investigate strategies to improve HPV vaccine attitudes and intent of HIs in the U.S.A. by developing and testing the effectiveness of video-based interventions. Methods: This study employed a two-phase mixed-methods approach. In the first phase, focus groups with new and established HIs explored perspectives, concerns about HPV vaccination, types of information to include in a video intervention, and how an effective intervention should be designed. Findings from the focus groups guided the creation of seven short educational videos, including a summary video and a testimonial-based video, addressing key questions about HPV and its vaccine. The second phase, which involved a nationwide survey of 1500 Spanish-speaking HIs, revealed a significant change in overall HPV vaccine attitude generally, and a significant increase in both HPV vaccine intent and attitudes among parents of unvaccinated children. Results: Regression analysis revealed general vaccine attitudes (β = 0.620, *p* < 0.001), English proficiency (β = 0.066, *p* = 0.01), and gender (β = −0.072, *p* = 0.002), as significant predictors of attitudinal changes. Notably, females exhibited less favorable post-intervention attitudes compared to males. Additionally, perceived care from video creators was a strong predictor of normalized gains in vaccine attitudes (β = 0.270, *p* < 0.001). Video content effectiveness varied; the video addressing vaccine side effects demonstrated the highest impact on attitude improvement. Testimonials and the summary video were also effective in fostering positive changes in attitudes. Despite differences in trust levels between new and established immigrants, both groups valued culturally tailored, Spanish-language information from credible sources. Conclusion: Addressing language and cultural barriers can improve trust in healthcare interventions among Hispanic immigrants in the U.S.A. Public health initiatives should consider these factors to more effectively reduce HPV vaccine hesitancy in this population.

## 1. Introduction

The introduction of vaccines has revolutionized public health and played a crucial role in improving child health by greatly decreasing mortality and disease rates [1,2,3]. A 2014 study estimated that without mass vaccination, diseases such as measles, mumps, rubella, tetanus, diphtheria, pertussis, polio, hepatitis B, varicella, and HPV would have caused around 20 million infections and 12,000 deaths or permanent disabilities [2]. Additionally, in 2019, routine childhood immunizations in the U.S. prevented over 24 million cases of vaccine-preventable diseases among children under 10 [3]. However, despite this robust success, a recent increase of vaccine hesitancy in the United States has hampered this progress.

Vaccine hesitancy, defined as the reluctance or delay in receiving vaccines despite their availability [4], has contributed to the persistence of vaccine-preventable diseases in the United States [5]. Childhood vaccination coverage in the U.S.A. dropped from 72.7% in 2015 to 70.4% in 2017 [6]. During the coronavirus disease 2019 (COVID-19) pandemic, early flu vaccination rates among children also decreased, falling from 34.2% in 2018 to 29.7% in 2020 [6]. This increased incidence of vaccine hesitancy is particularly concerning in the case of preventable diseases such as Human Papilloma Virus (HPV).

HPV is the most common sexually transmitted infection, affecting 80% of sexually active adults [7]. It is responsible for 70–75% of cervical cancer cases (the fourth most common cancer in women) [8], causing approximately 10,800 diagnoses each year in the U.S.A. [9]. To mitigate the impact of this issue, the HPV vaccine has been widely recommended for adolescents, with current guidelines advocating routine vaccination at ages 11–12 [10]. The HPV vaccine has been shown to significantly reduce HPV and cervical cancer rates [11,12], and multiple reviews have further advocated its safety and efficacy [13,14]. However, HPV vaccination rates in the U.S.A. remain lower compared to other routine adolescent vaccines. As of 2023, about 77% of adolescents aged 13–17 years had received at least one dose of the HPV vaccine, yet only 61% had received the full recommended series. Vaccination coverage has reduced in recent years, with uptake consistently falling behind other childhood immunizations, particularly among certain demographic groups [10]. For example, HPV vaccine uptake is remains unusually low compared to other vaccines, particularly among Hispanic immigrants (HIs) in the United States [15,16].

Hispanics are the largest and fastest-growing racial/ethnic minority in the United States, comprising 18% of the population [17]. However, while White American populations have experienced a decline in HPV rates, rates remain disproportionately high among Hispanics [18]. This is due to many factors, including lower screening rates [19], lower knowledge of HPV’s link to cervical cancer [20], more difficult healthcare access, and issues with immigration status [21], among other factors. One prevalent factor is a lower rate of HPV vaccine uptake [22]. This trend, combined with the significant presence of the Hispanic population in the U.S., strongly supports making increased HPV vaccine uptake a public health priority.

The gap in HPV vaccine uptake among Hispanic immigrants is not without cause. Spanish-speaking individuals in linguistically isolated areas face greater immunization disparities [23,24]. Additional research highlights the impact of the COVID-19 pandemic, vaccine safety concerns, limited health education, and lower levels of acculturation as barriers to vaccination [25,26]. Given the diversity within HI communities, pinpointing the exact reasons for vaccine hesitancy remains challenging, and several studies have called for further investigation [27,28,29].

This study used Spanish-language focus groups and surveys to explore factors influencing HPV vaccination decisions within the HI community, using the results to develop educational interventions. The focus group questions were based on a structural equation modeling (SEM) framework from a prior study by Larson et al. [24], which found a positive correlation between HPV vaccination and religious beliefs, trust in authorities, and reliance on home remedies, and that reducing language barriers strengthened patient–doctor trust, leading to more positive attitudes toward the HPV vaccine. To address language barriers, the intervention materials developed in this study were entirely in Spanish.

The Health Belief Model (HBM) and Cultural-Centric Narrative Theory were used in the conceptualization and design of our intervention. HBM is a psychological framework designed to explain and predict health-related behaviors, particularly the uptake of health services. This model was developed in the 1950s by several U.S.A. public health researchers [30,31]. This model theorizes that health behaviors are influenced by six key constructs: (1) the belief that one may acquire a disease or harmful state (perceived susceptibility), (2) the belief in the seriousness of potential consequences (perceived severity), (3) the belief in the advantages of taking action to reduce risk (perceived benefits), (4) the belief in the costs or obstacles to taking action (perceived barriers), (5) the triggers that prompt the need to take action (cues to action), and (6) the confidence in one’s ability to successfully perform a certain behavior (self-efficacy) [32].

Numerous studies have shown the usefulness of the HBM as a framework in health promotion, disease prevention, and vaccination intervention programs [33,34,35,36]. In the context of the HPV vaccine, studies have shown that HBM-based interventions positively influence knowledge, attitudes, and vaccine intent among college-aged males [37] and adolescents [38]. Additionally, this model has been used effectively to develop and evaluate interventions aimed at increasing HPV vaccination rates among Hispanic populations [39]. Therefore, we hypothesized that effective health communication strategies would highlight the severity of a threat, address perceived barriers, and emphasize the effectiveness of recommended HPV-protective behaviors.

The Cultural-Centric Narrative Theory explains the importance of culturally grounded narratives in promoting healthy habits and health communication. According to this theory, the effectiveness of health interventions improves significantly when participants can relate to the content on a personal level. Additionally, it provides a framework for the design of culturally adapted messages to shape health behaviors in specific audiences [40]. Based on this theory, we hypothesized that culturally appropriate interventions in Spanish would be more relevant for our participants.

Videos have been found to effectively convey information through visual and auditory cues [41]. Therefore, we predicted that our educational intervention would be most effective in the form of short videos, which addressed barriers, benefits, self-efficacy, and threats associated with HPV in an effective manner.

## 2. Materials and Methods

This project consisted of two phases. First, we conducted two focus groups with local HI communities to understand their perspectives and concerns about the HPV vaccine. We used the findings from these to create the educational video intervention. Second, we administered a nationwide survey to evaluate the effectiveness of these video interventions among Spanish-speaking HIs across the U.S.A.

### 2.1. Ethical Approval

The study was approved by the Institutional Review Board of the primary author’s institution, protocol number (IRB2023-392). Informed consent was obtained electronically from all participants before they began the survey.

### 2.2. Phase 1a: Focus Groups

Our focus groups had a combined representation of 13 HIs. The first focus group, conducted on 31 January 2024, at the Alliance Community Service building in Salt Lake City, comprised 5 “new” HIs (those who had lived in the U.S.A. for less than 2 years). Two participants were from Venezuela, one was from Chile, one was from Ecuador, and one was from Colombia.

The second focus group, conducted on 1 February 2024, at the same location, involved eight established HIs (those in the country for more than two years). Seven participants were from Mexico and the remaining one was from Venezuela. All participants provided consent for their focus group participation and for their comments to be recorded.

The focus group discussions were centered on participants’ perspectives and concerns about HPV and its vaccine, as well as preferred formats of educational materials that could potentially influence their vaccine attitudes. Table 1 outlines the 4 main sections and associated questions for the focus group.

Both focus groups were audio-recorded, and the recordings were transcribed verbatim (in Spanish) using Sonix transcription software (Premium version). The Spanish transcripts were then translated into English using a DeepL translator (Pro Starter). Three bilingual researchers meticulously reviewed both the English and Spanish texts for accuracy.

Inductive content analysis [42] was used to create emergent codes from the focus group responses. Coding was done by the principal investigator, two graduate researchers, and five undergraduate researchers. Codes were collaboratively and iteratively refined and then organized into themes using inductive coding. The following themes emerged from our focus groups: the importance of vaccination, concerns about vaccines, preferred information format (e.g., video), trust in information sources, familiarity with the HPV vaccine, and a need for clear and engaging content.

### 2.3. Phase 1b: Video Creation

Based on the results of the focus groups concerning format, actors, and information preferences, a series of six short (under 3.5 min), engaging videos using information from credible sources, such as the Centers for Disease Control and Prevention (CDC), were created by the research group. Members of the research group were recruited from an education research lab that specializes in science communication surrounding evolutionary biology and vaccination, as well as in the creation of high-quality educational videos (such as in the following studies: [43,44]).

These videos addressed key questions about HPV and its vaccine, including the following: What is HPV? How is HPV transmitted? How do I prevent HPV? Where does the vaccine come from? At what age should I vaccinate my children against HPV? Are there side effects of getting vaccinated? We also included a somewhat longer (just under 8 min) seventh video showing testimonials from three parents who support vaccination for their children and provide their perspectives. The videos featured diverse perspectives from real individuals, including doctors, scientists, parents, and HPV vaccine recipients, based on the focus group respondents’ preference for who they trusted. Visual aids, such as animations and graphics, were used to enhance understanding. The content was designed to be easily comprehensible and provide reliable information about the HPV vaccine, including its efficacy, potential side effects, and impact on various age groups, particularly in the context of early relationships and sexual health (see Appendix A for a full description of each video in Appendix A).

### 2.4. Phase 2: Assessing Effectiveness of Interventional Videos 

A survey containing the video interventions was distributed to a representative sample of 1500 Hispanic immigrants across the United States through Qualtrics (Provo, UT, USA). Eligibility criteria for survey inclusion were ethnicity (Hispanic, Latina, or Spanish origin), Spanish language fluency, a country of birth outside the USA, being a parent, and being over 18 years of age. Additionally, respondents were stratified by income and education to be representative of the USA Census. The survey was validated for face validity by a virologist, a specialist in biological education and religious influences, and a specialist in Spanish culture. Most survey questions were validated from a previous survey, which measured vaccine attitudes, HPV knowledge, English language use, language barriers, trust in medicine/institutions, financial barriers, and demographics [24]. Three native Spanish speakers validated the accuracy of the Spanish translation of each survey item. Confirmatory Factor Analysis (CFA) was used to confirm reliability and validity after data collection. We performed CFA on each latent variable’s measurement model. We used several fit indices, including the Tucker–Lewis index (TLI), the comparative fit index (CFI), the root mean square error of approximation (RMSEA), and the standardized root mean square residual (SRMSR). Mplus software, version 8 estimator was used to perform the CFA. The overall fit statistics confirmed there was no need for any major changes (see Appendix A).

The survey was composed of three major parts: pre-intervention, intervention, and post-intervention. Pre-intervention questions measured the following latent factors: general vaccine attitudes (e.g., “Vaccines are more beneficial than harmful”), HPV vaccine uptake (e.g., “Have you been vaccinated against HPV?”), attitudes towards the HPV vaccine (e.g., “I believe it is safe to give the HPV vaccine to my children”), knowledge of HPV (e.g., “HPV is a potentially life-threatening infection”), openness/comfort/fear (e.g., “I might consider vaccinating my children if I learn more about it”), and English proficiency (e.g., “How comfortable do you feel speaking English at work?”) (see Appendix A for the full survey in Spanish and English).

The intervention included a 5 min summary video that took highlights from all six videos addressing each key question. After viewing this, respondents were asked if they would like more information and given the option to watch one full video from the list of key questions. Additionally, they could choose to watch a testimonial video from real parents. Or they could choose to watch no additional videos. Based on their choice, respondents viewed the full video on that topic.

Post-intervention questions included all the pre-survey questions (excluding demographics) along with care level (e.g., “The makers of these videos have my best interests at heart”) and video-specific feedback (e.g., “I feel that these videos provided reliable information about the HPV vaccine”).

### 2.5. Statistical Analysis

Frequency distributions were computed for the demographic variables. Paired sample *t*-tests and ANOVA were used to measure participants’ HPV vaccine attitudes and intent pre- and post-intervention. Normality and homogeneity of variance were checked with both ANOVA and *t*-tests. Multiple Linear Regression analysis was conducted to identify significant predictors of vaccine attitude changes. Additionally, normalized gains were calculated to assess the relative improvement in attitudes. ANOVA was used to compare the effectiveness of the different video types on attitude changes. All statistical analysis was completed via Statistical Package for Social Sciences (SPSS, Version 29.0.1.0). A result was statistically significant if the *p*-value was less than or equal to 0.05. 

## 3. Results 

### 3.1. Phase 1a: Focus Group Results

We found several commonalities but a few differences between the two immigrant populations. Among new immigrants, some emphasized the importance of vaccination from birth, while others voiced concerns particularly about the COVID-19 vaccine. They preferred shorter, topic-specific videos that addressed various aspects of the HPV vaccine and information in Spanish. Interestingly, they placed a high level of trust in information from the U.S. government. One participant stated, “So [if] it says, ‘This comes from the government’. You’re kind of reassured too, because it’s already something that’s certified and approved. This information is approved and endorsed”.

Contrastingly, established immigrants did not trust information from the U.S. government for vaccination decisions and exhibited mistrust similar to native citizens [43]. One participant commented when asked if information from the Centers for Disease Control would be helpful, “More than the government, because the government is very much a liar”. Some expressed concerns about the effectiveness of new vaccines, while others shared positive experiences, particularly in managing health conditions. They preferred videos featuring real people, such as healthcare professionals and individuals with personal experiences, and emphasized the need for clear and engaging content. They expressed concern about the importance of addressing issues related to vaccine efficacy and potential side effects, particularly those related to sexual health.

Both groups discussed the likelihood of vaccinating themselves and their family against HPV if their concerns were addressed regarding safety, effectiveness, potential side effects, and the importance of providing accurate and trustworthy information about the HPV vaccine from credible sources such as the Centers for Disease Control and Prevention (CDC). Additionally, both groups emphasized the importance of evidence-based information and personal experiences, the need for clear and comprehensive educational resources, and a preference for multiple short videos that cover different aspects of the HPV vaccine.

### 3.2. Phase 1b: Video Creation

We identified main themes expressed in focus groups and used these to create associated video content. Table 2 shows the main themes accompanied by example quotes. It then outlines decisions we made for video content. Software tools that were used to create and edit the educational video interventions can be found in Appendix A.

### 3.3. Phase 2: Survey Results

#### 3.3.1. Sociodemographic Characteristics

The study population comprised 1188 participants, of whom 63.7% identified as female and 35.2% as male; 1.1% of participants did not provide gender data. The age distribution showed that the majority of participants fell within the 26–35 age range (32.8%), followed by 36–45 years (29.2%), with small proportions in the 18–25 (13.7%), 46–55 (11.4%), and over 55 (12.8%) age categories. The racial composition of the sample was predominantly Hispanic/Latino, constituting 74.1% of the total, followed by participants identifying as mixed race (14.6%) and White (8.9%), with minimal representation from Native American (0.7%), Black (0.8%), Asian (0.1%), and other races (0.8%). Furthermore, the data indicated that nearly three-quarters of the participants (73%) had resided in the United States for a period longer than five years, while 21.5% reported a residency of 1–5 years, and 5.4% had resided in the U.S. for less than one year.

The distribution of household income levels showed some variations, with 15.7% of the participants reporting annual incomes ranging from $76,000 to $100,000, and 13.7% of the participants having annual incomes above $100,000. Conversely, 22.5% of respondents reported annual incomes of $25,000 or less. The majority of participants reported having children (99.9%), with most households having two (33.9%) or three children (23.1%). The data on marital status showed that the majority of respondents were either married (58.8%) or in a partnership (16.0%), while a smaller proportion were single (12.5%), divorced (9.8%), or widowed (2.1%). Most participants indicated that a reasonable portion of their family resided in the United States (60.3%), while 32.9% reported family living in their home country, and 6.4% had family living in a country other than the U.S.A. or their home country.

The data showed variations in educational attainment, with 33.9% of the sample having completed high school, 23.1% reporting some college or vocational training, and 19.7% holding a bachelor’s degree. Additionally, 8.1% had obtained an advanced degree, such as a master’s or doctoral degree, while 7.2% had not completed high school (see Appendix A for detailed demographic data in Appendix A).

#### 3.3.2. Attitudinal Data

We used a paired sample *t*-test to assess the impact of the intervention on the uptake of and attitudes towards the HPV vaccine. We demonstrated a positive, but not significant, trend in overall HPV vaccine intent (willingness of participants to vaccinate themselves or their children against HPV) from pre-intervention (*M* = 14.73, *SD* = 3.89) to post-intervention (*M* = 14.89, *SD* = 3.68), *t*(1154) = −1.94, *p* = 0.053. We demonstrated a statistically significant change in overall HPV vaccine attitudes (personal beliefs, perceptions, and values regarding the safety, need, protective benefits, and what they think about vaccination as a way of keeping children healthy) from pre-intervention (*M* = 14.90, *SD* = 4.18) to post-intervention (*M* = 16.13, *SD* = 4.05), *t*(1159) = −13.61, *p* < 0.001; see Table 3 and Figure 1. 

In a subgroup analysis of only parents whose children were unvaccinated, we found a significant increase in both HPV vaccine intent from pre-intervention (*M* = 13.80, *SD* = 3.81) to post-intervention (*M* = 14.25, *SD* = 3.70), *t*(879) = −4.73, *p* < 0.001, and in HPV vaccine attitudes from pre-intervention (*M* = 14.27, *SD* = 4.25) to post-intervention (*M* = 15.67, *SD* = 4.19), *t*(884) = 13.18, *p* < 0.001; see Figure 1.

The regression analysis of post-intervention HPV vaccine attitudes revealed three significant predictors: general vaccine attitudes pre-intervention, English proficiency, and gender. General vaccine attitudes emerged as a strong predictor (β = 0.620, *p* < 0.001), indicating that individuals with more positive general vaccine attitudes were more likely to have improved HPV vaccine attitudes post-intervention. English proficiency also showed a significant positive relationship (β = 0.066, *p* = 0.01), suggesting that higher proficiency in English is associated with better post-intervention HPV vaccine attitudes. Additionally, gender was a significant predictor (β = −0.072, *p* = 0.002), with the negative coefficient indicating that females had less favorable post-intervention HPV vaccine attitudes than males. English proficiency also significantly predicted initial attitudes toward the HPV vaccine (F (1,1157) = 159.71, *p* < 0.001; β = 0.348, *p* < 0.001). However, English proficiency did not predict gains in attitudes.

To determine whether the perceived level of care (i.e., how much respondents felt cared for by the video makers) predicted gains (normalized) in HPV vaccine attitudes, we conducted an independent sample *t*-test to compare the care levels between new (less than two years in the country) and established (more than two years in the country) immigrants. The results indicated that the perceived care levels were identical between the two groups. Further regression analysis confirmed that the level of care was a significant predictor of gains in HPV vaccine attitudes (F (1,1000) = 78.78, *p* < 0.001; β = 0.270, *p* < 0.001), with higher perceived care predicting greater improvements in HPV vaccine attitudes; see Figure 2.

We conducted an ANOVA to examine the effect of watching additional videos on normalized gains in HPV vaccine attitudes. It should be noted that the proportion of participants that watched each additional video is much smaller than the full sample. The results revealed significant differences in normalized gains in vaccine attitudes depending on the type of video watched (F (6,1003) = 3.57, *p* = 0.002). Following the significant ANOVA results, we conducted post hoc analyses to further explore the impact of different videos on normalized gains in HPV vaccine attitudes. Video 5 (“Are there side effects of getting vaccinated?”) had the highest normalized gain (0.301), indicating it was the most effective in positively changing attitudes toward HPV vaccination. Video 6 (“Testimonials”) and choosing no additional video to watch also showed substantial normalized gains (0.271 and 0.268, respectively), suggesting that testimonials and the main video itself were effective. Conversely, Video 4 (“At what age should I vaccinate my children against HPV?”) had a negligible normalized gain (−0.042), indicating it may have been less effective in positively changing attitudes.

About a quarter of participants (*n* = 277, 18% of the total sample) chose not to watch an additional video. The most popular videos were Video 1 (“What is HPV”), with 183 participants, followed by Video 2 (“How do I prevent HPV?”), with 168 participants. Video 3 (“Where does the vaccine come from?”) was watched by 117 participants, Video 5 (“Are there side effects of getting vaccinated?”) was watched by 105 participants, and Video 4 (“At what age should I vaccinate my children against HPV?”) was watched by 92 individuals. Video 6 (“Testimonials”) was the least popular, with only 68 participants selecting it.

The care level varied slightly across the different videos but was relatively high overall. Video 1 (“What is HPV”) and Video 6 (“Testimonials”) had the highest care levels, with scores of 21.53 and 21.44, respectively. Video 3 (“Where does the vaccine come from?”) had the lowest care level, with a score of 17.07.

## 4. Discussion

As we consider how to deliver health messaging to a variety of populations, it is imperative that we understand our audience and tailor our messaging to be most resonant and, therefore, effective. In this study, our focus was on communicating information about the HPV vaccine to Hispanic immigrant populations. We took a two-pronged approach, first engaging with both new and established immigrant populations to learn about their beliefs, hesitancies, and preferences surrounding vaccine communication, and then tailoring educational interventions to match. We then tested these interventions and found them to be successful in increasing vaccine attitudes and intents surrounding HPV vaccination.

### 4.1. Community Engagement

Both new and established Hispanic immigrant populations emphasized the importance of unbiased, evidence-based, and credible information in communicating about HPV vaccination. Interestingly, in an analysis of informational materials available for HPV, Brandt et al. [41] found that most material was insufficient for a general audience, with the majority written at too high a reading level, most being unsuitable for the intended purposes, and nearly two-thirds having missing information. Previous research by Freed et al. has suggested that Hispanic populations tend to be more concerned about adverse side effects surrounding vaccinations than their White and Black counterparts [44], suggesting that they especially would benefit from better, more accessible information. However, it is important to note that this research also found that Hispanic respondents were more likely to follow recommendations of their doctors and less likely to refuse a vaccine than both White and Black respondents.

Both focus groups emphasized the desire to hear from people who would have presumed knowledge and training in the medical and research fields. This emphasizes the importance of the messenger. A recent CDC report emphasized the importance of “trusted messengers” in communicating about vaccines (in their case, the emphasis was on the COVID-19 and flu vaccines, but presumably this would also apply to HPV) [45]. Also intriguing was the emphasis participants put on hearing first-hand information from parents who have vaccinated their children and potentially from patients who have experienced HPV infection. Previous work on vaccine hesitancy has shown that appealing to personal experiences can significantly improve vaccine attitudes, especially when the experiences include anecdotes from those who have suffered consequences of vaccine-preventable diseases [46].

Interestingly, new immigrants had more favorable views toward information from the U.S. government than established immigrants. This seems contrary to some recent research that would suggest trust should be low among immigrants. In an interview of immigrant parents who had both foreign-born and US-born children, DeRose [47] found that trust issues were highly prevalent based on their experience with disparate care between foreign- and USA-born children when seeking vaccines. Similarly, Galletly et al. [48] found that Hispanic immigrants participating in focus groups were clearly aware of an image of them being a “burden” on the U.S. healthcare system, many of whom came to the erroneous conclusion that healthcare was unavailable to them. Even US citizens of Hispanic origin tend to fear that the U.S. government may scrutinize their immigration status if they seek out healthcare [49]. In our study, however, these new immigrants had been in the USA less than five years and, therefore, may have had limited experience with the U.S. healthcare system and government services.

Lastly, in asking focus group participants about formatting for educational materials, all participants emphasized the need for short, informational videos, but also the need for interest-capturing features and animations to clarify concepts. With the advent of social media platforms predominated by short, entertaining clips or quotes (e.g., TikTok, Instagram, X (formerly Twitter), it is not surprising that this is a preferred method of communication. A recent study on preference for online content confirmed that respondents prefer engaging, short-format content that can easily be viewed on a mobile device during short breaks in a busy day [50].

### 4.2. Translating Community Feedback into Successful Educational Interventions

Informed by community feedback, we created short educational videos that covered a variety of topics related to HPV vaccination. We also utilized multiple sources, including a university researcher, government authorities (the CDC), and first-person testimonials. Additionally, we listened carefully to format preferences expressed in focus groups and designed short, engaging, whiteboard-animated videos along with first-person dialogue. Lastly, we made all videos in the Spanish language. In a study utilizing the Health Information National Trends Survey from 2008 to 2014, researchers found that among Hispanic individuals, native language preference (i.e., Spanish) was a major deterrent to seeking out health information [51]. Confirming this, Chu et al. [52] found that Spanish-language respondents were more likely than English- or Chinese-speaking respondents to struggle to understand health information when seeking it out. Given this research, we chose to present all health information in the Spanish language.

We tested the effectiveness of these videos on changing attitudes about vaccination through a survey administered to Hispanic Immigrants across the country. We found that our intervention played a significant role in positively influencing vaccine attitudes, especially among parents with unvaccinated children. In two previous reviews of the literature on affecting change in HPV vaccine uptake, one analyzing the published literature from 2006 to 2015 [53] and the other from 2015 to 2018 [54], researchers found support for the use of educational interventions in improving attitudes. However, both reviews highlighted the lack of sufficient studies on this type of intervention, especially among ethnic minorities. This study adds to this literature, suggesting that educational interventions can improve vaccine attitudes among Hispanic minorities.

It is interesting to note that intent to vaccinate was not significantly changed by our intervention in the full population that included a large number of parents who had already vaccinated their children. This is likely caused by a “ceiling effect” where intent was already high in a good portion of the population. However, when investigating only parents who have not vaccinated their children for HPV, the intervention significantly increased their willingness to vaccinate. This indicates that educational interventions of this sort may be a good avenue to reach vaccine-hesitant populations. However, it is important to note that changes in willingness may not translate to actual vaccination or completion of the vaccination series. In fact, interventions leading to significant changes in initiating or completing the HPV series are few [54].

As has been seen in previous research [7,24], a respondent’s attitude toward vaccines in general is a significant predictor of attitudes toward the HPV vaccine. This suggests that interventions to improve vaccine attitudes can have an additive effect in improving vaccine uptake. However, researchers caution that a one-size-fits-all approach to vaccine communication may not be effective, given that individuals may use very different decision-making processes when deciding to vaccinate themselves or their children [55].

Our study revealed that English proficiency was a significant predictor of both pre- and post-intervention attitudes about the HPV vaccine. This is in line with previous research. Lee et al. [27] showed that English proficiency was significantly associated with awareness of the HPV vaccine. In regard to the COVID-19 vaccine, recent research has linked low English proficiency with a delay in receiving the vaccine and an increase in COVID-19-associated hospitalization and death [56]. Interestingly, however, with our intervention, English proficiency did not predict gains in attitudes. We suggest that by providing the intervention entirely in Spanish, we may be overcoming language barriers that may have inhibited information attainment, thereby allowing participants to fully engage in content and experience changes in attitude. Based on this, we would recommend that educational interventions designed for immigrant populations should be provided in native languages so as to reduce barriers to information.

Interestingly, we found that gender was a significant predictor of post HPV vaccine attitudes, with females demonstrating lower attitudes. Previous research has found that women are more likely to be concerned about general vaccine side effects and to espouse the belief that vaccines cause autism, leading to more rejection of vaccines [44]. However, Applewhite et al. [57] showed that when it comes to the flu vaccine, women are more likely to vaccinate than men. And it is similar for HPV vaccination, with women being far more likely than men to be familiar with the HPV vaccine [58,59], and women having a higher uptake than men [60,61].

Each video in our intervention presented different information about HPV vaccination. We found that the video discussing potential side effects of vaccination (e.g., fever, chills, malaise) coupled with reinforcing the potential benefits (e.g., avoiding HPV infection and potential cancer) was most effective in increasing respondent attitudes. Research shows that awareness of the experience of infection can lead to better attitudes toward vaccination [e.g., [46,62]]. We would recommend that educational interventions include information about potential inconveniences surrounding vaccination, but to weigh this against the significant prevention of serious disease that vaccines provide.

The second most effective video in changing attitudes was the testimonial video highlighting parents who have vaccinated their children. This supports recent research showing that testimonials about the COVID-19 vaccine (in this case, from local clinicians) was effective in increasing vaccine attitudes [63]. Similarly, Cawkwell and Oshinsky [64] consider storytelling as a potentially effective tool in communicating health information surrounding vaccines. Jennings et al. [65] found that messaging that contained both facts and narratives was most effective at increasing vaccine advocacy. Our intervention included both the composite informational video and the opportunity to watch narrative testimonials. We would suggest that future vaccine interventions consider including narrative voices along with the informational message. However, these results should be considered with caution given the much smaller sample size that viewed each additional video.

We were very interested in the degree to which the respondents felt cared for by the creators of the educational videos. According to the theory of planned behavior, an individual’s behavior is driven by their beliefs, their attitude towards a particular situation, what they think other people would expect of them, and how they feel in control of their own behavior [66]. This theory posits that engagement in specific health behaviors is contingent upon the belief that such behaviors will yield personally valuable outcomes [67]. Indeed, we found that the higher a respondent’s perceived care level, the more effective the video seems to be at changing their intentions and behavior. This aligns with the theory’s emphasis on the role of beliefs and attitudes in shaping intentions and behaviors. When respondents felt that the creators genuinely cared about their well-being, they were more likely to trust the information presented, engage with the content, and ultimately assimilate and consider the recommended health information.

Interestingly, respondents indicated different levels of perceived care based on the additional video they watched. The video that discussed straightforward information about the HPV pathogen and how it is transmitted had the highest level of perceived care. For this video, in particular, the narrator was a native Spanish speaker, so it is possible that participants might have made a stronger cultural connection to the message being shared in a familiar accent. The testimonial video had the second-highest perceived care level. Evidence supports that narrative communication has intrinsic emotional value and that this is important in health interventions, especially in the context of health and risk messaging where narratives address sensitive topics such as illness susceptibility and vulnerability [68]. Several factors can be associated with the emotional potency of narratives in health communication. According to Baesler and Burgoon [69], vivid language and imagery characteristics of narratives can create mental representations that evoke strong emotional responses in recipients. Volkman [70] confirms that the vividness of these mental images contributes to the overall emotional impact of the narrative, enhancing its persuasive potential in health interventions.

The video that discussed the development of the vaccine and highlighted the Pfizer pharmaceutical company had the lowest perceived care. It is unclear as to why, but we suspect it may have to do with the controversial messaging surrounding Pfizer in the wake of the COVID-19 vaccine [71]. Respondents exposed to such messaging may be inclined to distrust messaging highlighting this company.

### 4.3. Limitations of Our Study

While this study adds to our understanding of how best to communicate with Hispanic immigrant populations about vaccination, there are some limitations of which to take note. First, due to logistics, our focus groups were conducted both in a local area with immigrants from only certain Central and South American countries. So, caution should be taken when broadening implications to other Hispanic immigrant populations. Our survey was conducted nationwide, so we are more confident in the success of the intervention for a broader population. It is important to note, however, that we only measure intent to vaccinate, not actual vaccine uptake. This is often not equivalent to actual follow-through rates [54]. This may, in part, be due to social desirability, where survey participants provide responses that they believe are expected or more socially accepted rather than giving genuine opinions or true intentions [72]. Further research needs to be done on actual vaccination behaviors following interventions such as these. Another limitation, due to our method of surveying, is that participants only watched a summary video and then one other video. They were not exposed to all videos in the intervention, as this would require too much viewing time. However, we believe that if patients seek out information, they would be more likely to watch additional videos. Even with the summary video, however, the intervention was successful in increasing vaccine attitudes. We did not investigate why some participants chose not to watch an additional video. This limits our understanding of potential barriers to further engagement with contents of health educational interventions. Further studies should explore potential factors that influence participants’ willingness to engage with multiple components of an educational intervention.

### 4.4. Implications for Health Communication

Community engagement, one of the most important takeaways from this study, has proven to be a crucial strategy in understanding communication preferences and designing effective health-related interventions. By actively involving communities, researchers and health communicators can tailor health information that resonates deeply with the target population. This study emphasizes the importance of delivering health information in participants’ native language. This ensures comprehension and fosters a sense of cultural relevance. While English proficiency emerged as a predictor of attitudes, it was not a barrier to attitudinal shifts. This emphasizes the important role of native language communication in achieving meaningful behavioral change.

Additionally, participants rated videos high in terms of quality and level of care that they felt from the video producers. This suggests that culturally sensitive and empathetic health messages are essential for effective health promotion. Health communicators must prioritize reducing language barriers and integrating culturally relevant elements into their messaging to better engage Hispanic immigrant populations. This study emphasizes the need for unbiased, credible information in health interventions. The lack of accessible, high-quality resources for underserved communities calls for urgent efforts to produce culturally relevant and evidence-based health materials. Our study further emphasizes the need for unbiased, credible information transfer when communicating about health interventions.

### 4.5. Implications for Public Health Policy, Insurance, and Government

The results of our study indicate that addressing language and cultural barriers is relevant for public health initiatives targeted at reducing HPV vaccination disparities and, by extension, the incidence of HPV-related cancer among HIs in the USA. For healthcare policy makers, this suggests that prioritizing culturally appropriate educational materials and outreach campaigns (within broader cancer prevention initiatives) may lead to significant results. Since provider recommendation is a known driver of vaccine uptake, we recommend that healthcare policy makers prioritize training for healthcare professionals in cultural competence and communication [73,74]. The results of our study indicate that interventions fostering trust—particularly those perceived as caring and community-based—are effective. These findings should guide health policies for HPV vaccination and preventive care among immigrant and minority communities in the USA.

Given that data indicate insurance status is a key factor influencing HPV vaccine uptake among young adults in the USA. [75], we recommend that insurers invest in multilingual outreach campaigns to educate members about HPV vaccine benefits and coverage, particularly targeting parents of unvaccinated children.

Government agencies could also lead large-scale, culturally tailored public health campaigns using effective formats such as testimonial and short videos, as these were shown to effectively drive positive attitude change. Additionally, better tracking of vaccination rates by language and acculturation rates could help inform intervention strategies.

## Figures and Tables

**Figure 1 vaccines-13-00574-f001:**
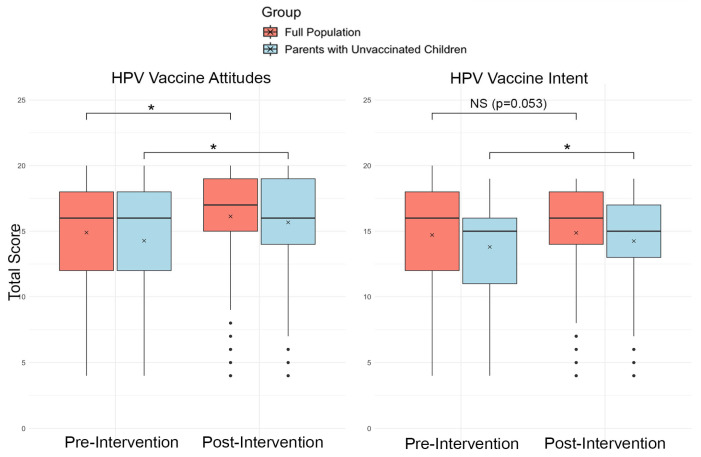
Changes in HPV vaccine attitudes and intent from pre-to-post intervention in the general population (*n* =1155) and parents of unvaccinated children (*n* = 885). Boxes represent the interquartile range (IQR), horizontal lines denote medians, the “x” denotes means, and whiskers represent 1.5 × IQR. Asterisks (*) indicate significant changes between pre- and post-intervention scores, as determined by paired *t*-test (*p* < 0.05).

**Figure 2 vaccines-13-00574-f002:**
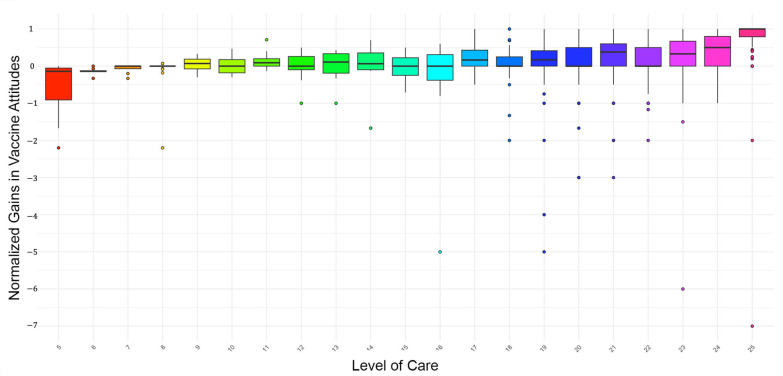
Level of care and corresponding normalized gains in HPV vaccine attitudes.

**Table 1 vaccines-13-00574-t001:** Focus group questions.

Section	Focus Group Questions
Vaccine Attitudes and Awareness	1. What are your general feelings about vaccines? 2. Are you familiar with the HPV Vaccine? How likely are you to get your kids vaccinated?
Trust and Concerns in Vaccine Communication	3. Who is the ideal person to communicate health information concerning vaccines? 4. What makes you more or less trusting of that individual? 5. Do you worry about providing documentation when you visit a healthcare provider? 6. Do you fear vaccines based on a lack of information? If so, what information is missing?
Specific Concerns and Influences on HPV Vaccination	7. Are there any concerns you have specific to HPV vaccination that we have not covered above? 8. How much do your religious views impact your decisions about the HPV vaccine? 9. Besides going to a medical doctor, who do you go to for treatments?
Effective Communication Strategies for HPV Vaccine Education	10. We are planning to produce educational materials (in the form of short videos) to help Spanish-speaking immigrants better understand the HPV vaccine. Based on what you have shared, we would like a little more feedback on what you think would be most effective for these videos. 11. Who should be the person speaking? 12. What kind of information would be helpful? 13. What format do you think is best (someone just relaying information to you vs. watching a conversation between a doctor and patient vs. a conversation between two parents)? 14. What makes you most comfortable listening to the information?

**Table 2 vaccines-13-00574-t002:** Emergent themes from focus group and corresponding video content decisions. (The focus group number is in parentheses; Group 1 = New immigrants, Group 2 = Established immigrants).

Emergent Theme	Example Quote
What Information is needed?	
Disease information	“And if there is a video on the complexity of the Papilloma issue, I think there are several factors that should be included, such as at least how it is acquired, who is directly affected, how to prevent it, what are the best, that is, the best resources to use for the vaccine”. (group 1)
Video Decision:	Included a video titled, “What is HPV?” that details the characteristics of the virus, how it is transmitted, and the consequences of infection.
Benefits/effectiveness of the vaccine	“Like for example…what percentage of effectiveness does it have [and] to know how much that vaccine affects them or how much that vaccine can help them or how much it can benefit them?” (group 2)
Video Decision:	Included a video titled, “How Do I Prevent HPV?” that explains how vaccines function to prevent disease, specifically highlighting the HPV vaccine. Additionally, we included a video titled, “Where did the Vaccine Come From?” that highlighted the long-term protection from and prevention of HPV-related cancers and consequences that the vaccine provides.
Side effects of the vaccine	“What are the effects? And after the vaccine? What is going to happen, tell us if the child is going to have a fever or something?” (group 1)
Video Decision:	Included a video titled, “Are there Side Effects of Getting Vaccinated?” that explains how safe the HPV vaccine is and what common side effects are seen.
Vaccine origin and manufacture	“How did it originate? How was it created? Who created it? Which laboratories?” (group 1)
Video Decision:	Included a video titled, “Where did the Vaccine Come From?” that highlighted both the approval process of the vaccine and safety statistics, as well as the long-term protection from and prevention of HPV-related cancers and consequences that the vaccine provides.
How and when to administer	“One question I have is what would be the right age to give the vaccine? Because we are talking about an age range that is possible, but as she explained to us it’s for a child of eight or nine years old. But we see that it is not necessary [for them] because they will not have the same concern we have about sexual relations. So what would be the right age to administer the vaccine?” (group 1)
Video Decision:	Included a video titled, “At What Age Should I Vaccinate My Children Against HPV?” that states the recommended age range and then explains why that age range is appropriate given the need to prevent infection prior to being sexually active.
Who should give the information?	
Medical Professionals/Research Scientists	“That you have the credentials to be able to prove what you’re saying…Because if they offer you a vaccine and they don’t know what it is, they haven’t studied it or they don’t know what it is, how are you going to trust that person? That’s what it’s like to have the basis, uh, to have studied. It doesn’t matter if it is a doctor or a nurse, but someone related to the field of the vaccine. Someone who knows how to clarify doubts”. (group 1)
Video Decision:	We chose to include an expert who is a university professor of microbiology to explain how HPV is transmitted.
Government/Center for Disease Control (CDC)	“As an example. This information comes from CDC”. (group 1)
Video Decision:	We chose to highlight the statement on the HPV vaccine directly from the Center for Disease Control and Prevention (CDC). We also highlighted the role of the US Food and Drug Administration (FDA) in the approval process of the vaccine.
Testimonials	“The experience of someone who’s been through it tells you about it, too. It helps, because you say hey, it’s real. It’s not nothing but a story. Like at the beginning of COVID when everybody started dying. Hey, let’s get vaccinated”. (group 1)
Video Decision:	We included a video, titled “Testimonials” highlighting three native Spanish-speaking women talking about their own experiences with the vaccine and their children.
Medical Professionals/Research Scientists	“That you have the credentials to be able to prove what you’re saying…Because if they offer you a vaccine and they don’t know what it is, they haven’t studied it or they don’t know what it is, how are you going to trust that person? That’s what it’s like to have the basis, uh, to have studied. It doesn’t matter if it is a doctor or a nurse, but someone related to the field of the vaccine. Someone who knows how to clarify doubts”. (group 1)
What format is appropriate?	
Engaging content	“Show pictures and make it didactic. How do you say, more fun? Fun is not just standing up and talking and giving a talk and that, because no, that’s not the idea. But doing, showing you the images, showing you the dialogues all of that kind of stuff, but talking to you directly”. (group 1)
Video Decision:	We chose to use a whiteboard animation application, “Video Scribe”, to create videos with animation, embedded videos of people, and both on-screen and voice-over text
Direct Communication	“Someone to talk to me directly, like this, to tell it all like it is. The information, the side effects”. (group 1)
Video Decision:	We used first-person format where all videos, and people in them, are talking directly to the viewer.
Short	“Imagine three min each with one for each point, it would be better”. Another speaker said, “Because I don’t have 20 min. Right, I don’t know about you guys, but we’re busy. We as dads don’t have time”. (both group 1)
Video Decision:	All videos (except “Testimonials”) were under 3 ½ min. The testimonial video was just under 8 min.
Engaging content	“Show pictures and make it didactic. How do you say, more fun? Fun is not just standing up and talking and giving a talk and that, because no, that’s not the idea. But doing, showing you the images, showing you the dialogues all of that kind of stuff, but talking to you directly”. (group 1)
Other:	
Language Barrier	“Sometimes we don’t take the interest, sometimes they put you a translator, but the translator doesn’t tell you things well as the doctor can tell you”. (group 2)
Video Decision:	All videos were filmed entirely in Spanish by native speakers or non-native speakers who have spent two years or more immersed in a Spanish-speaking environment. This includes all on-screen text, voice-over text, and person dialogue.

**Table 3 vaccines-13-00574-t003:** Pre- and post-intervention means, standard deviations, t-values, and *p*-values for HPV vaccine uptake and attitudes.

	Means	Standard Deviations	*t*	*p*
Pre-HPV vaccine intent	14.73	3.89	−1.94	0.053
Post-HPV vaccine intent	14.89	3.68		
Pre-HPV vaccine attitude	14.90	4.18	−13.61	<0.001
Post-HPV vaccine attitude	16.13	4.05		

## Data Availability

Data will be made available on BYU Scholars Archive upon acceptance of the manuscript. This will be replaced with an Accession Number.

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
