# Peer review of "Intervention Strategies to Overcome HPV Vaccine Hesitancy Among Hispanic Immigrants in the USA: A Video-Based Approach"

_vaccines, 2025, doi:10.3390/vaccines13060574_

Round 1

Reviewer 1 Report

Comments and Suggestions for Authors

This is a study that evaluated the impact of educational video on HPV vaccine uptake by Hispanics in the US.

  1. General comment:

While assessing the impact of educational videos on participants' opinion of the vaccine is helpful, it would have been more meaningful if the participants who changed their attitudes were actually offered the vaccine afterwards to see if they indeed would take it and just agree to take it without taking the action to do so. This is a common limitation of survey study known as "social desirability bias." This should be acknowledged by the authors in the limitations section.

Other issues that need to be addressed:

  1. The results part of the abstract needs the addition of the numerical results of the regression analysis to help justify the strength of correlation of each listed factor wit vaccine uptake.
  2. The abstract lacks a conclusion part.
  3. The introduction is extremely long. I suggest the authors reduce its word count to fill a max of a page and a half as possible.
  4. Line 99: Remove the subheading. Introductions typically don't have subheadings.
  5. Line 181: To help with study replicability, please list all the resources you used in preparing the content of the videos and add their citations. This could be either in text, in a table, or in a supplementary material along with links to these resources. This way other authors who wish to follow your steps can find the same resources that you used in your videos.
  6. Line 185: What do you mean by [e.g., 46,47]?
  7. How long is each video? You mentioned that the summary video was 5-minute long, but what about the six educational videos.
  8. Line 232: The percentage of participants who didn't want to watch an additional video should be moved to the results. Also, did you investigate why weren't they interested?
  9. It's good that the authors assessed which videos had the greatest impact. However, the proportion of participants who watched those videos is small since watching an additional video was optional.
  10. Did the authors try to assess the impact of the number of watched videos on the attitude toward HPV vaccine just like they did by qualitatively assessing which videos had the greatest impact? This can give an idea on how powerful a single video could be.

Reviewer 2 Report

Comments and Suggestions for Authors
  1. Page 2, Line 62-68, would the authors please introduce the prevalence of HPV in USA and the rate of HPV vaccination in USA.
  2. The full spelling should be provided for “COVID-19” in its first place in Line 78.
  3. Page 6, Line 248, the p value for significant difference should be defined.
  4. In the text, they are same for “COVID-19” and “COVID”, “USA” and “US”? If yes, please unify.
  5. The citations should be formatted according to the guideline of Vaccines

Author Response

Thank you for your helpful feedback. You will find our responses adjacent to your comments in the table below:

Reviewer 2

Page 2, Line 62-68, would the authors please introduce the prevalence of HPV in USA and the rate of HPV vaccination in USA

Great suggestion! This has been addressed on page 2 in the updated copy on page 1, line 71-8 to say "HPV is the most common sexually transmitted infection, affecting 80% of sexually active adults(7). It is responsible for 70-75% of cervical cancer cases(the fourth most common cancer in women)(8), causing approximately 10,800 diagnoses each year in the USA(9). To mitigate the impact of this issue, the HPV vaccine has been widely recommended for adolescent, with current guidelines advocating routine vaccination at ages 11-12(10). The HPV vaccine has been shown to significantly reduce HPV and cervical cancer rates(11,12), and multiple reviews have further advocated its safety and efficacy(13,14). However, HPV vaccination rates in the USA remain lower compared to other routine adolescent vaccines. As of 2023, about 77% of adolescents aged 13-17 years had received at least one dose of the HPV vaccine, yet only 61% had received the full recommended series. Vaccination coverage has reduced in recent years, with uptake consistently falling behind other childhood immunizations particularly among certain demographic groups (10). For example, HPV vaccine uptake is remains unusually low compared to other vaccines, particularly among Hispanic immigrants (HI) in the United States(15,16)". 

Reviewer 2

The full spelling should be provided for “COVID-19” in its first place in Line 78.

Thank you for pointing out this. This has been resolved on page 2, line 67 as "coronavirus disease 2019 (COVID-19)".

Reviewer 2

Page 6, Line 248, the p value for significant difference should be defined

Thank you for your comment. To address this, we have now defined the p-value threshold for statistical significance in the revised copy (page 7, line 274).Specifically, a result was considered statistically significant if the p-value was less than or equal to 0.05.

Reviewer 2

In the text, they are same for “COVID-19” and “COVID”, “USA” and “US”? If yes, please unify

Thank you for pointing this out. Yes! we were referring to COVID-19 instead of COVID and USA instead of US. We have made these uniform in our revised text.

Reviewer 2

The citations should be formatted according to the guideline of  Vaccines

Thank for bringing this up. We have resolved this issue. Check page 20, line 722-900.

Reviewer 3 Report

Comments and Suggestions for Authors

Hispanics have a higher rate of cervical cancer for a number of reasons such as, lower screening rates, limited access to treatment, and sociocultural influences. There is little discussion of this in the introduction and must be addressed. 

On line 96 the authors talk about language barriers as an issue but fail to address that in their research design. Why? What are the reasons that this was not included? If there aren't any solid reasons then the research design is flawed. 

The same for the lower screening rates, limited access to treatment, and sociocultural influences. 

What was the regression method used? Little to know discussion of the type of regression analysis. It would seem to me that a Logit model would be appropriate here and that could address the issues above if they were asked in the survey. 

In the discussion, there is a presentation of the results. However, what do these results mean for healthcare policy and what do they mean for insurance companies and government for example.

Author Response

Thank you for your helpful feedback. You will find our responses adjacent to your comments in the table below:

Reviewer 3

Hispanics have a higher rate of cervical cancer for a number of reasons such as, lower screening rates, limited access to treatment, and sociocultural influences. There is little discussion of this in the introduction and must be addressed

Excellent point. We have revised this paragraph in the introduction (page 2, line 87-92 ) to say "However, while white American populations have experienced a decline in HPV rates, rates remain disproportionately high among Hispanics(18). This is due to many factors including lower screening rates(19)lower knowledge of HPV’s link to cervical cancer(20), more difficult healthcare access and issues with immigration status(21) among other factors. One prevalent factor is a lower rate of HPV vaccine uptake(22)".

Reviewer 3

On line 96 the authors talk about language barriers as an issue but fail to address that in their research design. Why? What are the reasons that this was not included? If there aren't any solid reasons then the research design is flawed

Thank your for this question. We discussed language barriers in the introduction because this was one of the motivations for the current study. We tried to eliminate the language barrier by providing all intervention materials in their native language of Spanish. We have tried to clarify this a little better by revising the statement on page 3, line 119 that follows to say, "To address language barriers, the intervention materials developed in this study was entirely in Spanish."

Reviewer 3

The same for the lower screening rates, limited access to treatment, and sociocultural influences.

This is a great suggestion. Unfortunately, lower screening rates, limited access to treatment, and sociocultural influences, while having been shown as barriers, were not the focus of our intervention. Our intervention was focused primarily on the language and knowledge barriers.

Reviewer 3

What was the regression method used? Little to know discussion of the type of regression analysis. It would seem to me that a Logit model would be appropriate here and that could address the issues above if they were asked in the survey.

Thank you for pointing this out. We used multiple linear regression, rather than a logit model because our outcome variable was continuous (vaccine attitudes). We did not measure whether or not they actually got vaccinated, so a logit model would not work for these data. However, we see the confusion and have clarified in our methods that the regression was multiple linear on page 7, line 269.

Reviewer 3

In the discussion, there is a presentation of the results. However, what do these results mean for healthcare policy and what do they mean for insurance companies and government for example

Thank for your thoughtful feedback. We acknowledge that this important aspect was not fully captured in our discussion. In response to your comment, we have detailed the adjustments made to address this point, including a new subsection titled" Implications for Public Health Policy, Insurance, and Government intervention"(page 18, line 639-659) in our revised copy.

Round 2

Reviewer 1 Report

Comments and Suggestions for Authors

I appreciate the efforts made by the authors to address all the comments and make adjustments to the manuscript. I have no further comments.

Reviewer 2 Report

Comments and Suggestions for Authors

The manuscript can be accepted in present form, however, proofreading is needed before publication.

Reviewer 3 Report

Comments and Suggestions for Authors

There are some minor editing details to take care of, not grammatical in nature. 

Otherwise, I am satisfied with the changes to the paper that the authors have made and thank them for working hard to improve the paper.